# Quantifying and preventing *Plasmodium vivax* recurrences in primaquine-untreated pregnant women: An observational and modeling study in Brazil

**Rodrigo M. Corder**[1]*, **Antonio C. P. de Lima**[2], **David S. Khoury**[3], **Steffen S. Docken**[3], **Miles P. Davenport**[3], **Marcelo U. Ferreira**[1]*

1 Department of Parasitology, Institute of Biomedical Sciences, University of São Paulo, São Paulo, Brazil,
2 Department of Statistics, Institute of Mathematics and Statistics, University of São Paulo, São Paulo, Brazil,
3 Kirby Institute for Infection and Immunity, University of New South Wales, Sidney, Australia

* rodrigo.corder@usp.br (RMC); muferrei@usp.br (MUF)

**Data Availability Statement:** All relevant data are within the manuscript and its Supporting Information files.

## Abstract

Each year, 4.3 million pregnant women are exposed to malaria risk in Latin America and the Caribbean. *Plasmodium vivax* causes 76% of the regional malaria burden and appears to be less affected than *P. falciparum* by current elimination efforts. This is in part due to the parasite's ability to stay dormant in the liver and originate relapses within months after a single mosquito inoculation. Primaquine (PQ) is routinely combined with chloroquine (CQ) or other schizontocidal drugs to supress *P. vivax* relapses and reduce the risk of late blood-stage recrudescences of parasites with low-grade CQ resistance. However, PQ is contraindicated for pregnant women, who remain at increased risk of repeated infections following CQ-only treatment. Here we apply a mathematical model to time-to-recurrence data from Juruá Valley, Brazil's main malaria transmission hotspot, to quantify the extra burden of parasite recurrences attributable to PQ ineligibility in pregnant women. The model accounts for competing risks, since relapses and late recrudescences (that may be at least partially prevented by PQ) and new infections (that are not affected by PQ use) all contribute to recurrences. We compare recurrence rates observed after primary *P. vivax* infections in 158 pregnant women treated with CQ only and 316 *P. vivax* infections in non-pregnant control women, matched for age, date of infection, and place of residence, who were administered a standard CQ-PQ combination. We estimate that, once infected with *P. vivax*, 23% of the pregnant women have one or more vivax malaria recurrences over the next 12 weeks; 86% of these early *P. vivax* recurrences are attributable to relapses or late recrudescences, rather than new infections that could be prevented by reducing malaria exposure during pregnancy. Model simulations indicate that weekly CQ chemoprophylaxis extending over 4 to 12 weeks, starting after the first vivax malaria episode diagnosed in pregnancy, might reduce the risk of *P. vivax* recurrences over the next 12 months by 20% to 65%. We conclude that post-treatment CQ prophylaxis could be further explored as a measure to prevent vivax malaria recurrences in pregnancy and avert their adverse effects on maternal and neonatal health.

**Funding:** This work was supported by the Fundação de Amparo à Pesquisa do Estado de São Paulo (FAPESP; http://www.fapesp.br/en/), Brazil (grants 2013/21728-2, 2016/18740-9 and 2017/50292-9), the University of New South Wales (https://www.unsw.edu.au/), Australia, the National Health and Medical Research Council (https://www.nhmrc.gov.au/), Australia, and by the National Institute of Allergy and Infectious Diseases, National Institutes of Health, (https://www.niaid.nih.gov/), United States of America (grant U19 AI089681). RMC receives a doctoral fellowship from the Conselho Nacional de Desenvolvimento Científico e Tecnológico (CNPq; http://cnpq.br/), which also provides a senior research scholarship to MUF. The funders had no role in study design, data collection and analysis, decision to publish, or preparation of the manuscript.

**Competing interests:** The authors have declared that no competing interests exist.

## Author summary

*Plasmodium vivax*, a malaria parasite that can stay dormant in the liver and originate relapses within months after a single mosquito inoculation, causes 76% of the malaria burden in Latin America. Pregnant women are ineligible for primaquine (PQ), the only currently available drug that is able to prevent *P. vivax* relapses. Here we apply a mathematical model to real-life data from Brazil's main malaria transmission hotspot and estimate that, once infected with *P. vivax*, 23% of the pregnant women will have one or more vivax malaria recurrences over the next 12 weeks. Significantly, 86% of these early *P. vivax* recurrences are attributable to relapses or late recrudescences, which could be prevented by PQ administration. Repeated vivax malaria infections during pregnancy are associated with adverse effects on maternal and neonatal health. We show that weekly CQ chemoprophylaxis extending over 4 to 12 weeks, starting after the first vivax malaria episode diagnosed in pregnancy, might reduce the risk of *P. vivax* recurrences over the next 12 months by 20% to 65%, and should be investigated as a measure to lower the burden of repeated vivax malaria during pregnancy.

## Introduction

*Plasmodium vivax* causes over 14 million clinical malaria cases each year worldwide and appears to be less affected than *P. falciparum* by current elimination efforts in Latin America and the Asia-Pacific Region [1]. One distinctive feature of *P. vivax* associated with its increased resilience is the ability to stay in the liver as a dormant stage, the hypnozoite, following a primary infection. As a result, repeated episodes of blood-stage infection, known as relapses, may originate over the next weeks or months from hypnozoites reactivating at different time points following a single mosquito inoculation [2].

Radical cure of *P. vivax* infections entails the use of primaquine (PQ), the only hypnozoitocidal agent currently available in endemic settings, combined with one or more schizontocidal drugs, such as chloroquine (CQ) or artemisinin derivatives. Interestingly, PQ can also synergize the blood schizontocidal effect of CQ and thereby reduce the risk of recrudescence of partially CQ-resistant blood-stage parasites [3, 4]. However, not all patients can benefit from PQ use. First, some subjects cannot properly metabolize PQ, an inactive pro-drug. Because biotransformation mediated by the cytochrome P450 (CYP) isoenzyme CYP2D6 is required for PQ antirelapse activity [5], patients carrying low-activity CYP2D6 variants may present relapses despite PQ treatment [6]. Second, not all patients can take PQ. This drug cannot be administered to subjects with severe forms of glucose-6-phosphate dehydrogenase (G6PD) deficiency, since they may develop life-threatening hemolysis following treatment. Moreover, PQ is contraindicated for pregnant and lactating women and children below six months of age, because of the risk of hemolysis in fetuses and infants of unknown G6PD status [7]. Finally, not all patients adhere to the 7-day PQ regimen commonly used in Latin America [8, 9].

Despite the major recent progress towards malaria elimination in Latin America and the Caribbean, with an overall 62% decrease in incidence between 2000 and 2015, focal transmission persists mostly across the Amazon Basin [10]. An estimated 4.3 million Latin American women are at risk of malaria during pregnancy each year [11] and between 6,000 and 9,000 laboratory-confirmed malaria cases, over two thirds of them due to *P. vivax*, are notified yearly among pregnant women in Brazil [12].

The present study addresses the impact of PQ ineligibility on *P. vivax* malaria burden in pregnant women using real-life data from Juruá Valley, Brazil's main malaria transmission hotspot [13]. In this setting, repeated *P. vivax* infections during pregnancy have been associated with significantly reduced maternal hemoglobin concentrations and impaired fetal growth [14, 15]. Here we fit a mathematical model to time-to-recurrence data to quantify recurrence rates and estimate the period during which a pregnant woman is at increased risk of relapsing after her primary, PQ-untreated *P. vivax* infection, compared to PQ-treated non-pregnant control women. Moreover, we estimate the potential effectiveness of post-treatment chemoprophylaxis with weekly CQ to reduce the burden of repeated *P. vivax* infections during pregnancy in malaria-exposed populations.

## Material and methods

### Ethics statement

The study protocol was approved by the Institutional Review Board of the Institute of Biomedical Sciences, University of São Paulo, Brazil (CEPH-ICB 1368/17).

### Study area and population

We analyze malaria case notifications from the municipalities of Cruzeiro do Sul (07˚39' 54"S, 72˚39' 01"W) and Mâncio Lima (07˚36' 51"S, 72˚53' 45"W), both situated in the upper Juruá Valley, next to the border between Brazil and Peru. With <0.5% of the Brazilian Amazon's population (82,075 inhabitants in Cruzeiro do Sul and 17,545 inhabitants in Mâncio Lima), these municipalities together account for 15.7% of 194,000 cases recorded countrywide in 2017 [16]. With a typical equatorial humid climate, Juruá Valley receives most rainfall between November and April, but malaria transmission occurs year-round and is mainly due to *Anopheles darlingi*. The annual parasite incidences (API) have been estimated in 2016 at 436.4 cases per 1,000 inhabitants in Mâncio Lima, the country's highest API, and 231.9 cases per 1,000 inhabitants in Cruzeiro do Sul, the country's fourth highest API. Approximately 89.2% of the local infections are due to *P. vivax* and 10.8% due to *P. falciparum*, *P. malariae* or mixed species [16].

### Study design and data collection

This retrospective observational cohort study primarily aimed to compare the time to *P. vivax* recurrence following treatment, over 12 months of follow-up, in two groups of vivax malaria patients living in Cruzeiro do Sul or Mâncio Lima: (i) PQ-untreated cases, defined as women who reported to be pregnant and had a *P. vivax* infection treated with CQ alone (total dose, 25 mg of base/kg over 3 days), and (ii) non-pregnant control women presenting with an uncomplicated vivax malaria episode at the baseline and treated with the standard CQ-PQ regimen (CQ total dose, 25 mg of base/kg over 3 days; PQ, 0.5 mg of base/kg/day for 7 days) currently recommended in Brazil [17]. Parasite recurrences may be due to relapses (hypnozoite-derived infections), late recrudescences (due to CQ failure to clear partially resistant blood stages beyond day 28 of treatment) or new infections (originating from a new infectious mosquito bite). We assume that adding PQ to the treatment reduces the risk of both relapses and late recrudescences, but does not affect the risk of new infections. We do not attempt to estimate separately the anti-relapse and anti-recrudescence efficacy of PQ. Pregnant women receiving a weekly prophylactic CQ dose of 300 mg over 12 weeks following CQ-only treatment or until delivery, a regimen that was recommended in Brazil at the time of the study (2014–16) to treat *P. vivax* malaria recurrences observed during pregnancy [17] but was rarely prescribed in the study site, were not eligible for inclusion. In fact, only one patient was excluded on this basis.

To identify cases and controls and find repeated malaria episodes following treatment, we retrieved from the Malaria Epidemiological Surveillance and Information System (SIVEP) database of the Ministry of Health of Brazil (http://200.214.130.44/sivep_malaria/) 68,869 electronic records of malaria episodes that were laboratory-confirmed in the study sites from 1 January 2014 through 30 September 2016. Access to this database has been granted by the Ministry of Health of Brazil to study the burden of malaria in pregnancy in our study site [14]. Because malaria is a notifiable disease in Brazil and diagnosis is not offered by local private laboratories and antimalarials cannot be purchased in drugstores, we assume that virtually all malaria episodes in these municipalities were diagnosed and treated in public facilities and notified to the Ministry of Health. Giemsa-stained thick blood smears routinely have at least 100 fields routinely examined for malaria parasites under 1000× magnification by experienced local microscopists. We recorded information on patients′ age, pregnancy status, place and date of diagnosis, parasite species, and treatment administered at the time of the baseline infection and subsequent malaria episodes.

Over 90% of *P. vivax* relapses in Brazil occur within the first 6 months after the primary infection [18–20]. We thus used a 6-month window to minimize the risk that the baseline episodes were actually relapses or late recurrences of a recent *P. vivax* infection, instead of being new infections. This is crucial because only infections originating from mosquito bites will generate hypnozoites and expose patients to the risk of relapses over the next months. To this end, we excluded from the analysis all pregnant and control women who had vivax malaria cases diagnosed within 6 months before the baseline episode and those with a primary vivax infection between January and June 2014, the first six months of follow-up, since the 6-month window before the baseline episode could not be examined.

To minimize the confounding effects on malaria risk of age-related immunity, seasonality and spatial differences in transmission intensity, two controls were matched for every case according to: (i) subjects′ area of residence, using the malaria clinic providing diagnosis as a proxy, (ii) age (±5 years), and (iii) date of diagnosis (±30 days). The matching process considered the 6-month window mentioned above; individuals with vivax episodes were eligible only if they had no malaria episode diagnosed in the previous 6 months from their respective baseline episode. Study participants were considered to be malaria-unsusceptible between days 1 and 28 after the 3-day CQ course was started, due to the prolonged post-treatment prophylactic effect of CQ. Indeed, day-28 (early) recrudescences following CQ-alone or CQ-PQ regimens remain very infrequent in our study site [21, 22].

Because our database comprised malaria case records rather than unique individuals, the same subject could have multiple records if one or more recurrences were diagnosed after the baseline infection. Since the database did not include unique patient identifiers, we used the patient's name, the patient's mother's name and the date of birth to identify repeated vivax malaria episodes in the same subjects, either cases or controls [23]. To compare names, we used the Jaro-Winkler string distance [24] as implemented in the *R* software package *stringdist* [25], which partially accounts for typographical variation in letter strings such as given names or surnames. Comparisons were limited to pairs of individuals with a maximum age difference of 2 years. We next used the nearest neighbours' approach [26] implemented in the *R* software package *MatchIt* to find two suitable controls for each case. Starting with the baseline case (episode 0) in a pregnant woman, the nearest neighbours' approach finds the two most similar controls to the baseline case, among all vivax malaria episodes diagnosed in the same malaria clinic in non-pregnant women of about the same age (± 5 years) within a maximum interval of 30 days. We excluded pregnant women for whom we failed to find two matched controls.

## Data analysis

We primarily aimed to compare cases and controls according to the time between the baseline episode and the first *P. vivax* recurrence ($t_{0-1}$) observed over the next 12 months. A secondary analysis considered the time between the first and the second recurrences ($t_{1-2}$). The study outcome (failure) was defined as any recurrent episode of *P. vivax* parasitemia recorded in the database, irrespective of parasite density or symptoms, that was diagnosed in study participants by thick-smear microscopy until the end of follow-up (30 September, 2016). Participants with incident *P. falciparum* infection during the follow-up were censored at the time of diagnosis.

We first performed standard survival analysis and used log-rank tests to compare $t_{0-1}$ and $t_{1-2}$ in PQ-untreated pregnant women and PQ-treated non-pregnant control women. We note, however, that competing risks should be accounted for in this time-to-recurrence analysis, because recurrences may be due to either relapses/recrudescences (that may be at least partially prevented by PQ) or new infections (that are not affected by PQ use). To deal with this fact, we considered two recurrence dynamics: (i) a fast dynamics driven mostly by relapses or late recrudescences and (ii) a slow dynamics mostly driven by new infections. Because we excluded subjects with one or more vivax malaria episodes within 6 months before the baseline episode, we assume that the study population is essentially free of hypnozoites at baseline. Moreover, cases and controls are exposed to a similar risk of new infections. Note that the 6-month window also excluded subjects who are at an exceptionally high risk of repeated infections. Therefore, we assume that the hazard of new infections is constant and equal in both groups and we model the time to recurrence as an exponential distribution. This assumption is supported by our analysis of the time elapsed between the baseline episode and the most recent previous vivax malaria episode recorded in cases and controls (S1 Text). With respect to the fast dynamics, we also considered constant hazard for relapses and late recurrences because we assume that these events are randomly distributed over time.

The mathematical formulation is as follows. Regarding the first *P. vivax* recurrence, we consider that a proportion $p_p$ of the cases (pregnant women) are susceptible to either relapses or recrudescences (rate $\beta_r$) and new infections (rate $\beta_n$), characterizing a typical competing risk problem. The remaining proportion $1 - p_p$ is susceptible only to new infections (rate $\beta_n$). The proportion of subjects who remain uninfected at a given time t (survival function), with a 28-day delay due to the prolonged post-treatment prophylactic effect of CQ, is indicated by (1).

$$S_p(t;\ (p_p,\ \beta_r,\ \beta_n)) = \begin{cases} 1 & ,\ t \le 28 \\ p_p \cdot e^{-(\beta_r+\beta_n)\cdot(t-28)} + (1-p_p) \cdot e^{-(\beta_n)\cdot(t-28)} & ,\ t > 28 \end{cases} \quad (1)$$

The same formulation is applied to the control group. Following the baseline episode, a proportion $p_c$ of the control group is susceptible to either relapses/recrudescences or new infections with the same rates as the cases (that is, with rates $\beta_r$ and $\beta_n$). However, we assume that a $p_c \ne p_p$, because relapses and late recrudescences are partially prevented by PQ [3, 4]. The remaining proportion of the control group $1 - p_c$ is considered to be susceptible only to new infections. Therefore, the proportion of control women who remain uninfected at a given time t (survival function) is indicated by (2).

$$S_c(t;\ (p_c,\ \beta_r,\ \beta_n)) = \begin{cases} 1 & ,\ t \le 28 \\ p_c \cdot e^{-(\beta_r+\beta_n)\cdot(t-28)} + (1-p_c) \cdot e^{-(\beta_n)\cdot(t-28)} & ,\ t > 28 \end{cases} \quad (2)$$

## Model fitting

In order to fit the model to empirical data, let Y be the time to event (clinical malaria episode) with survival distribution function $S(\cdot, \boldsymbol{\theta})$ and density function $f(\cdot, \boldsymbol{\theta})$, where $\boldsymbol{\theta}$ is a parameter vector ($\boldsymbol{\theta} = (p, \beta_r, \beta_n)$). Let Z be the time to a censoring event (*P. falciparum* episode or the end of follow-up) such that the observable time variable (X) is the minimum between Y and Z, and δ, an indicator variable denoting whether a time of malaria episode or a time of censoring was observed. The actual data to be observed for each subject may then be represented by

$$X = \min(Y, Z) \text{ and } \delta = I(X = Y).$$

Assuming independent and non-informative censoring, we compute the likelihood for both cases and controls based on all N individuals for inference on $\boldsymbol{\theta}$ as follows

$$l(\boldsymbol{\theta}|(\mathbf{X}, \boldsymbol{\delta})) = \prod_{i=1}^{N} [f(X_i, \boldsymbol{\theta})]^{\delta_i} \cdot [S(X_i, \boldsymbol{\theta})]^{1-\delta_i} \tag{3}$$

with $\mathbf{X} = (X_1, \ldots, X_N)$ and $\boldsymbol{\delta} = (\delta_1, \ldots, \delta_N)$. Because $\beta_r$ is assumed to be the same for both cases and controls, as well as $\beta_n$, we maximized the product $l_p \cdot l_c$ of the likelihood of pregnant and non-pregnant groups, respectively, over the parameters ($p_p$, $p_c$, $\beta_r$, $\beta_n$). To obtain the maximum likelihood estimates and 95% credible intervals for each parameter, we used Markov Chain Monte Carlo (MCMC) for continuous random vectors using a Metropolis algorithm implemented in the *R* software package *mcmc* [27] considering $10^6$ iterations.

## Results

### Study subjects

Fig 1 shows the study flow diagram considering the primary outcome. We identified 272 pregnant women in our database with one or more vivax malaria episodes treated with CQ alone between January 2014 and September 2016. None of them was prescribed weekly CQ prophylaxis after treatment. Of them, 72 pregnant women were excluded because they had a vivax malaria episode within 6 months before the baseline episode or might have had an episode during this period. An additional 42 pregnant women were excluded because we failed to find two suitable non-pregnant controls. The final analysis comprised 158 cases defined as PQ-untreated pregnant women (mean age, 22.8 years; range, 14–43 years) and 316 PQ-treated non-pregnant control women (mean age, 22.5 years; range, 10–42 years) (data available in S1 Data).

### Primaquine use and the first *Plasmodium vivax* recurrence

Overall, 59 (37.3%) of 158 pregnant women (51 were censored at day 365 and 48 were censored between days 28 and 365) and 80 (25.3%) of 316 matched controls (127 were censored at day 365 and 109 were censored between days 28 and 365) had one or more laboratory-confirmed *P. vivax* recurrences diagnosed within 12 months after their baseline vivax malaria episode. The average intervals to first recurrence, without considering censored individuals, were $t^P_{0-1} = 93.7$ days for pregnant cases (59 experiencing recurrences) and $t^c_{0-1} = 127.5$ days for non-pregnant controls (80 experiencing recurrences). The survival curves in Fig 2A show a significantly longer $t^c_{0-1}$ interval, compared to $t^P_{0-1}$, with a hazards ratio (HR) = 1.73; (95% CI, 1.23–2.42; Log-rank test, p = 0.001). We interpret the significant difference between $t^P_{0-1}$ and $t^c_{0-1}$ intervals as due to PQ-preventable relapses or late recrudescences of parasites with low-grade CQ resistance.

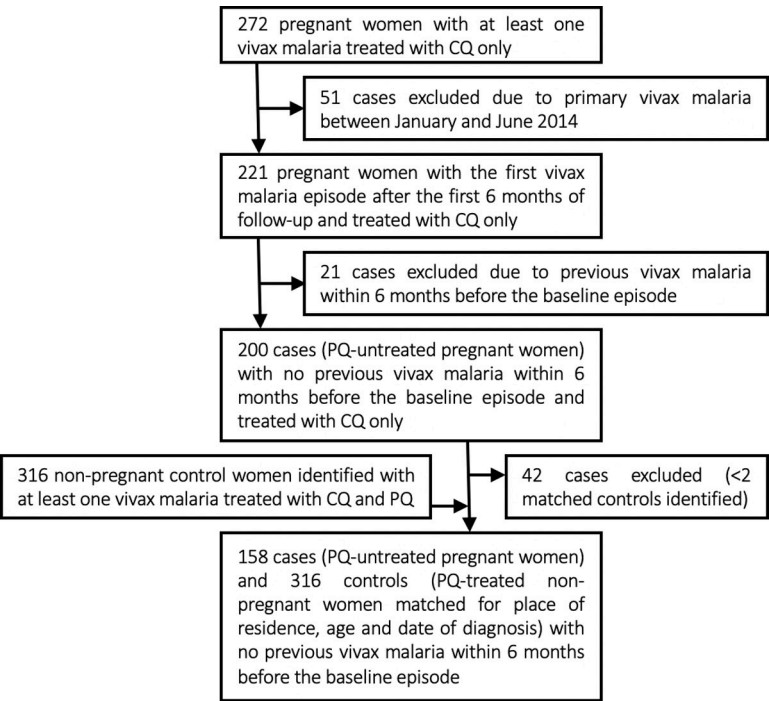

**Fig 1. Study flow diagram.** We identified 272 women who had one or more vivax malaria episodes during pregnancy diagnosed and treated with chloroquine alone between January 2014 and September 2016, and were not administered weekly chloroquine prophylaxis after treatment. Reasons for exclusion and the final number of subjects analyzed are indicated.

## Dynamics of the first *Plasmodium vivax* recurrence

Fitting our survival model (Eq 3) to the time to first recurrence data (using the MCMC approach described in the Materials and Methods section) yielded estimates of $\beta_r$, $\beta_n$, $p_c$ and $p_p$ (Fig 3). The fitted model estimated that 29.2% (95% CI: 19.8–40.2%) of pregnant ($\hat{p}_p$) and 11.7% (95% CI: 4.6–20.6%) of control women ($\hat{p}_c$) were susceptible to relapses or late recrudescences, with an average time to relapse/recrudescence of $\approx$46 days ($\hat{\beta}_r$ = 0.02157512 (95% CI: 0.01277316–0.03481974)). The average time to a new infection was estimated at $\approx$1440 days ($\hat{\beta}_n$ = 0.0006942688 (95% CI: 0.0003773581–0.0010288483)). We estimated that 28.3% of the pregnant women had a relapse or late recrudescence and 15.7% had a new infection causing their first vivax malaria recurrence within 12 months after the baseline episode. In contrast, 11.4% of the PQ-treated controls were estimated to have a relapse or late recrudescence and 18.8% were estimated to have a new infection causing their first recurrence over the same period of follow-up. Therefore, PQ-preventable relapses and late recrudescences were estimated to account for 64.4% and 37.7% of the first *P. vivax* recurrences observed in cases and controls, respectively, over 12 months of follow-up.

To better illustrate the dynamics of vivax malaria described by the model, we present the cumulative proportions of cases (Fig 4A) and controls (Fig 4B) that are expected to experience one of more recurrences over different follow-up periods. The model indicates that 35.3% of the cases and 20.1% of the controls will experience at least one vivax malaria recurrence within the first 6 months after treatment; most early recurrences are likely due to relapses or late recrudescences, rather than new infections, in both PQ-untreated cases (77.5%) and PQ-treated controls (54.5%). We next focus on the 12-week interval following

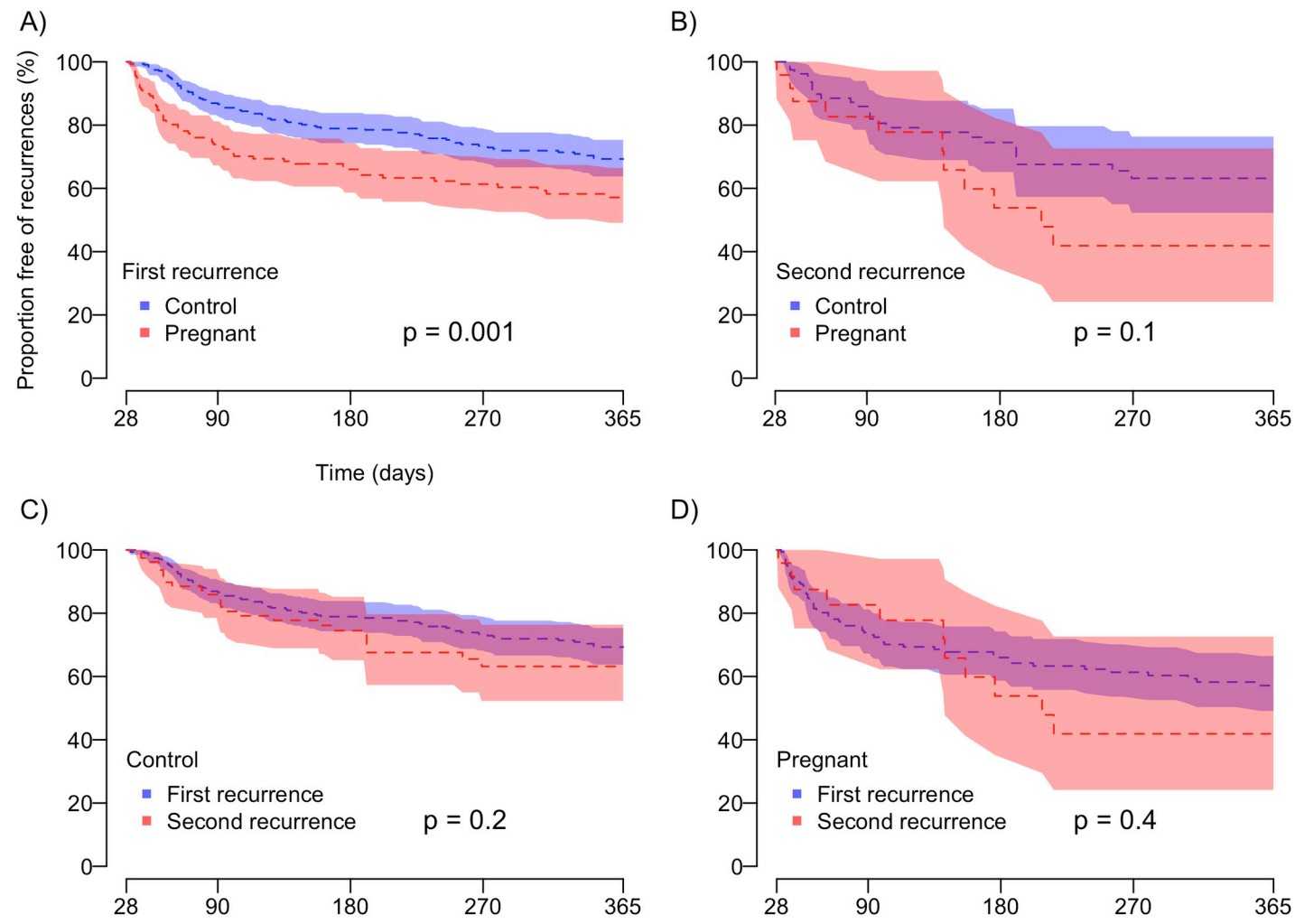

**Fig 2. Time-to-recurrence over 12 months of follow-up in PQ-untreated pregnant women and PQ-treated non-pregnant control experiencing a primary vivax malaria episode.** The shaded areas indicate the 95% confidence bands. PQ-untreated pregnant women were eligible for the second recurrence analysis if PQ was not given to treat the first episode. (A) Time to first vivax malaria recurrence, control vs. pregnant women; (B) Time to second vivax malaria recurrence, control vs. pregnant women; (C) Time to first recurrence vs. time to second recurrence, control women; (D) Time to first recurrence vs. time to second recurrence, pregnant women.

the primary vivax malaria episode, because this is the minimum duration of post-treatment CQ prophylaxis recommended in Brazil at the time of the study [17] but rarely prescribed in practice in the main malaria hotspot in this country. We estimate that 23.3% of cases and 11.6% of controls will experience one or more vivax malaria recurrences within 12 weeks after the primary episode, the vast majority of them (85.9% in cases and 69.2% in controls) being attributable to relapses or recrudescences. Therefore, a large proportion of repeated infections in pregnant women could be averted if CQ prophylaxis were administered following their first vivax malaria diagnosed during pregnancy, but the Ministry of Health of Brazil restricted the 12-week CQ prophylactic regimen following CQ-only treatment to *recurrent* (*but not primary*) vivax malaria episodes in pregnant women [17]. We further explore different scenarios of CQ prophylaxis after the primary vivax malaria episode in pregnancy in a separate section of this article.

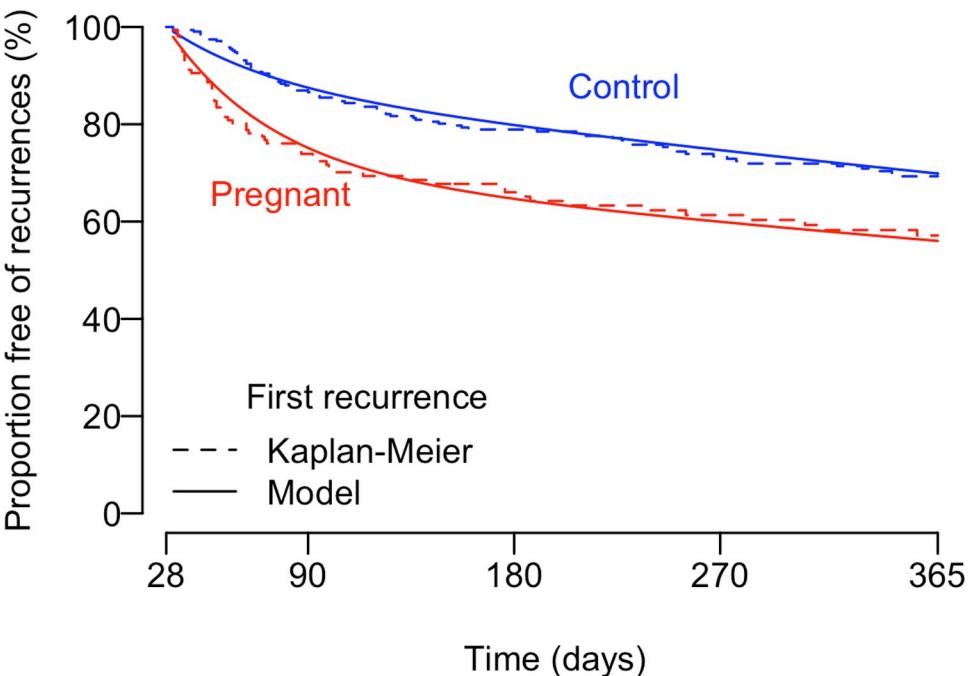

**Fig 3. Fitting of the mathematical model (Eqs 1 and 2) to the time to the first *P. vivax* recurrence over 1-year follow-up in PQ-untreated pregnant women and PQ-treated non-pregnant controls.**

## Subsequent recurrences

We next examined the intervals between the first and second recurrences in 24 cases ($t^P_{1-2}$) and 82 non-pregnant controls ($t^c_{1-2}$) (Fig 2B). Pregnant women were eligible for this analysis only if their first recurrence was treated with 3 days of CQ only–we therefore excluded women who were no longer pregnant at $t^P_1$, being thus given PQ, as well as those who received the

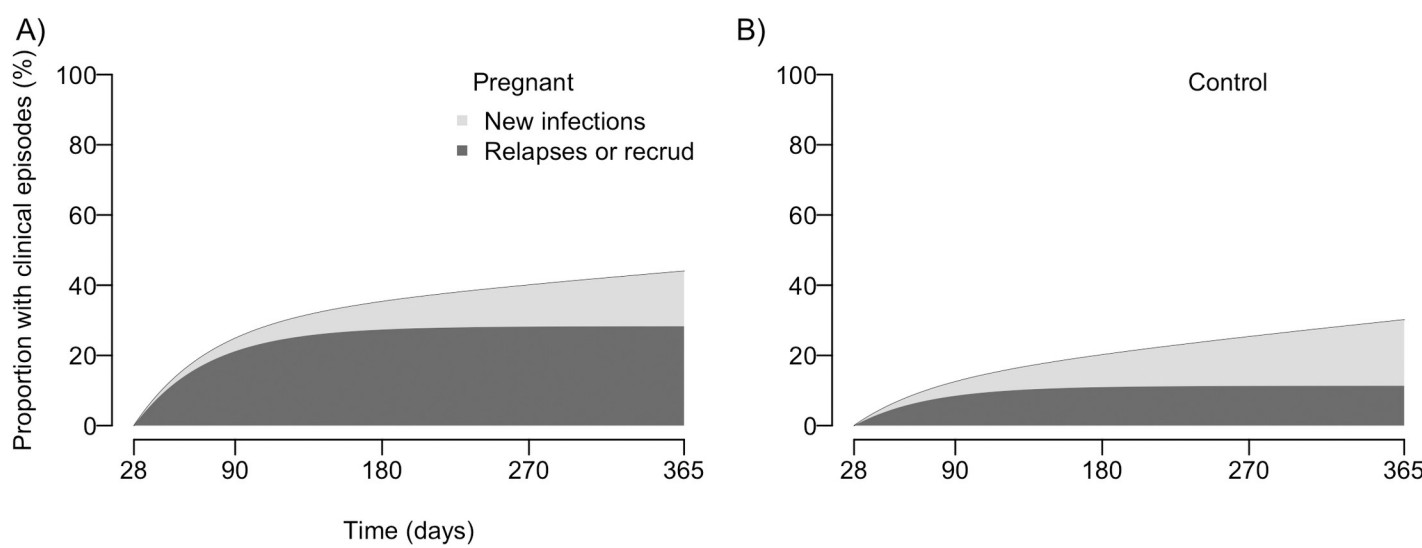

**Fig 4. Cumulative distribution function of first vivax malaria recurrences among PQ-untreated pregnant women and PQ-treated non-pregnant controls.** (A) First recurrence in pregnant women; (B) First recurrence in control women. The dark grey area in the graphs represents the proportion of subjects experiencing relapses/recrudescences and the light grey area represents the proportion of subjects experiencing new infections.

12-week post-treatment CQ prophylaxis. Overall, 11 (45.8%) of 24 pregnant women and 25 (30.5%) of 82 controls had repeated *P. vivax* episodes diagnosed up to 12 months after their first recurrence.

The average intervals between the first and the second recurrence, without considering censored individuals, were $t^P_{1-2}$ = 118.7 days among pregnant women (11 experiencing a second recurrence) and $t^c_{1-2}$ = 112.6 days among non-pregnant controls (25 experiencing a second recurrence). Figs 2C and 2D compare the times to the first ($t_{0-1}$) and second ($t_{1-2}$) recurrence in control and pregnant women, respectively. We found no significant difference between $t^c_{0-1}$ and $t^c_{1-2}$ (Log-rank test, p = 0.2; Fig 2C) or between $t^P_{0-1}$ and $t^P_{1-2}$ (Log-rank test, p = 0.4; Fig 2D).

After the first recurrence, some subjects may be free of hypnozoites while others may not be, especially PQ-untreated women. Therefore, we do not fit exponential distributions to the time to second recurrence.

### Preventing the first *Plasmodium vivax* recurrence with weekly chloroquine prophylaxis

There is no chemoprophylactic intervention that is widely recommended to reduce the burden of repeated vivax malaria infections in pregnancy in endemic settings worldwide. We thus examined whether weekly CQ chemoprophylaxis, introduced after the first primary, PQ-untreated vivax malaria episode during pregnancy, would reduce substantially the risk of subsequent parasite recurrences. We consider CQ regimens with a duration of 4, 8 or 12 weeks. Because compliance with unsupervised CQ prophylaxis may be relatively low and some degree of CQ resistance is possible, we assume that the regimens' efficiency in supressing *P. vivax* parasitemia while being administered, regardless of the origin of infecting parasites (hypnozoite- or mosquito bite-derived), ranges widely from 60% to 100%. We estimate that 4 weeks of post-treatment CQ prophylaxis administered to pregnant women in our study site would reduce by 20.4–34.0% the incidence of new blood-stage infection over the next 12 months (Fig 5). The vivax malaria incidence would decrease between 32.1% and 53.5% with 8 weeks of post-treatment prophylaxis and between 39.2% and 65.4% with 12 weeks of post-treatment prophylaxis after a primary PQ-untreated infection in pregnancy.

## Discussion

Until recently, the clinical and public health consequences of vivax malaria in pregnancy had been largely underestimated [28, 29]. Overall, *P. vivax* infections in pregnancy are associated with an increased prevalence and severity of maternal anemia, as well as with adverse birth outcomes such as low birth weight, preterm delivery, and miscarriage, both across Latin America [14, 15, 30] and globally [28–30]. These adverse effects are expected to be greater in areas of relatively low transmission, where there is little or no maternal immunity. Importantly, adverse birth outcomes are more common and severe among women experiencing repeated vivax malaria episodes during pregnancy, which are very common given their PQ-ineligibility, compared to mothers with a single antenatal infection [14, 31].

Since October 2012, the World Health Organization recommends that intermittent preventive treatment with sulfadoxine-pyrimethamine (IPTp-SP) be given at each antenatal care visit to prevent *P. falciparum* malaria in pregnancy in Africa. A minimum of three IPTp-SP doses, at least one month apart from each other, should be administered. However, as of 2017, the coverage of this regimen remains around 22%, on average, in the 39 African countries that have adopted the IPTp-SP policy [16]. No similar chemoprophylactic approach is currently recommended to prevent malaria in pregnancy in endemic settings outside Africa, where *P. vivax* is often the predominant infecting species, with the exception of Papua New Guinea.

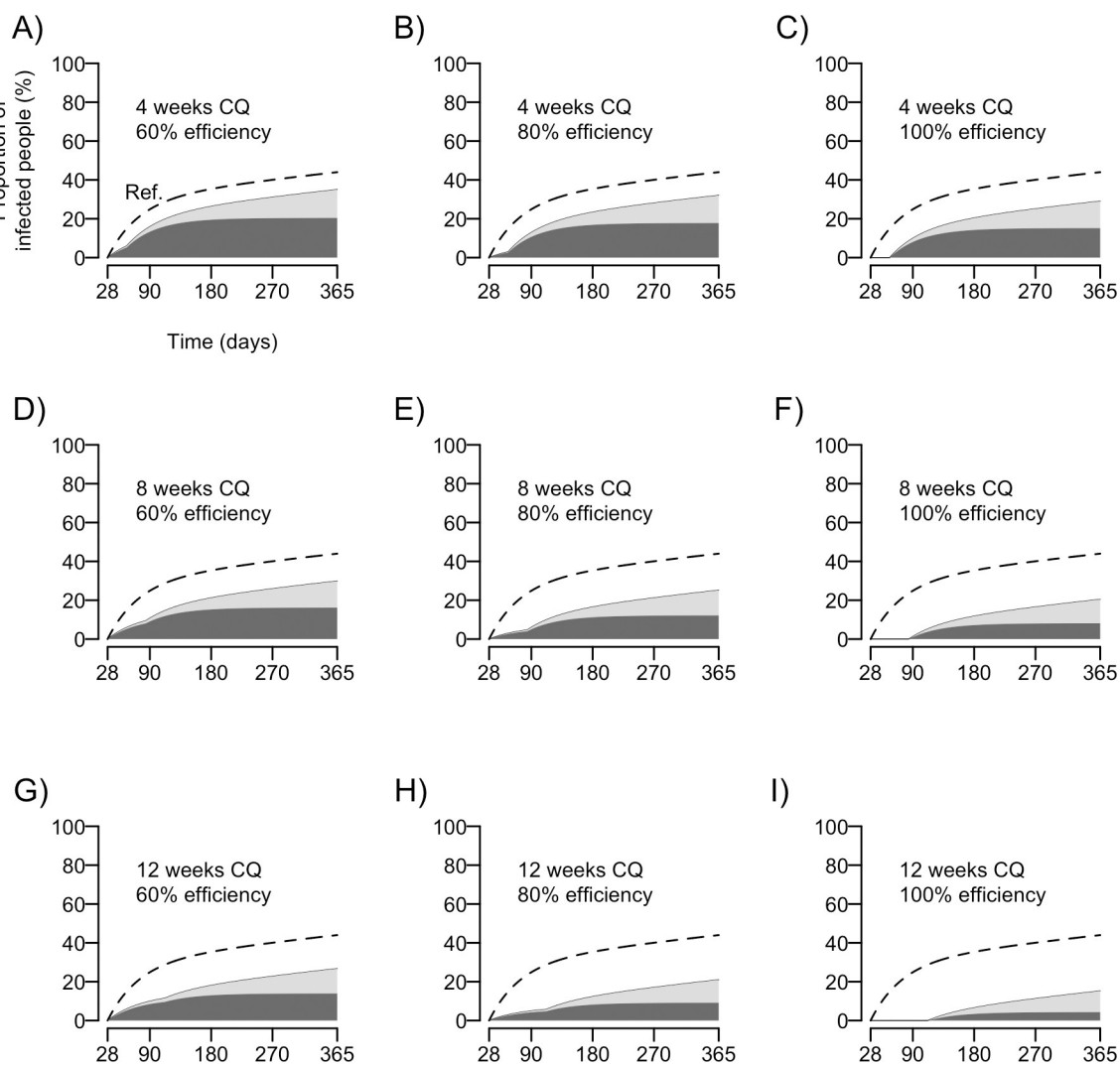

**Fig 5. Simulation of the potential effectiveness of post-treatment chemoprophylaxis with weekly chloroquine (CQ) to prevent *P. vivax* recurrences after a primary infection.** We considered scenarios with CQ prophylaxis being given over 4 (panels A, B and C), 8 (panels D, E and F) or 12 (panels G, H and I) weeks, being able to suppress 60% (panels A, D and G), 80% (panels B, E and H) or 100% (panels C, F and I) of blood-stage infections (regardless of its origin, whether hypnozoite- or mosquito bite-derived) during the period of administration. Dashed lines ("Ref.") represent the scenario without intervention; the dark grey area represents the cumulative proportion of subjects with *P. vivax* relapses/recrudescences over time and the light grey area represents the cumulative proportion of subjects with new *P. vivax* infections.

Moreover, experts convened by the World Health Organization in October 2017 concluded that current evidence does not support a change in the current recommendations on prevention, early diagnosis and treatment of vivax malaria in pregnancy followed by weekly CQ prophylaxis to reduce the risk of relapses [32]. In Brazil, repeated malaria screening with conventional microscopy or rapid diagnostic tests is recommended since 2006 as part of the standard antenatal care across the Amazon Basin, with prompt treatment of positive cases [12]. In Indonesia, routine malaria screening of pregnant women is also recommended since 2012, but testing is currently limited to the first antenatal care visit [33]. Malaria screening can be easily incorporated into the standard antenatal care in endemic settings, since blood samples are already collected from mothers for additional testing [34]. Nevertheless, only 6.8% of

the pregnant women attending antenatal care visits in three sentinel sites in the Brazilian Amazon were regularly tested for malaria in 2007–08 [35].

According to official malaria treatment guidelines in Brazil, the first *P. vivax* infection diagnosed during pregnancy should be treated with a CQ-only regimen over 3 days. At the time of the study (2014–16), the weekly prophylactic CQ dose following the regular 3-day regimen, to be administered over 12 weeks or until delivery, was limited to *P. vivax* recurrences, but not primary episodes, diagnosed during pregnancy [17]. However, this post-treatment prophylactic regimen was rarely prescribed in the main endemic setting of Brazil, leaving women at increased risk of repeated infections during the course of pregnancy. The reasons why weekly CQ prophylaxis remained neglected in our field site remain unclear. We speculate that the requirement for previous vivax malaria episodes during the course of the same pregnancy to introduce the prophylactic regimen may have drastically limited its use, since the attending physician does not have prompt access to this information. Importantly, however, the most recent malaria treatment guidelines of the Ministry of Health of Brazil, published in January 2020, now recommend the early introduction of weekly CQ prophylaxis, since the first vivax malaria episode diagnosed during the pregnancy, to be extended until delivery [36].

We estimate that 23% of the *P. vivax*-infected pregnant women in Juruá Valley present, over the next 12 weeks, one or more vivax malaria recurrences that could be potentially suppressed by post-treatment CQ prophylaxis starting after the primary infection. Approximately 86% of these early *P. vivax* recurrences (within 12 weeks of the baseline episode) are estimated to result from relapses or late recrudescences, rather than new infections. These findings indicate that measures to reduce malaria exposure in pregnant women, such as insecticide-treated bednet distribution during antenatal care visits, would prevent only a small proportion of these repeated malaria episodes. In contrast, our modeling simulations suggest that CQ chemoprophylaxis following the primary *P. vivax* infection in pregnancy might suppress between 20% (4-week CQ prophylaxis with 60% efficiency) and 65% (12-week CQ prophylaxis with 100% efficiency) of the first blood-stage recurrences experienced over the next 12 months, with major potential implications for mothers' and neonates' health.

We note that the modeling approach described in this article can be readily applied to different scenarios, such as clinical trials [37–39] and prolonged antimalarial stock outages [40], to estimate how many recurrences can be prevented by PQ or tafenoquine (TQ) in the presence of competing risks originating from new infections, provided that PQ- and TQ-untreated comparison groups are available. Importantly, our analysis does not assume complete relapse suppression by treatment, but rather estimates the proportion of infections described by the fast dynamics, consistent with real-life scenarios with partially efficacious drugs and relatively poor adherence of patients. However, we suggest that time-to-recurrence studies should be carefully designed in order to (i) minimize the risk that the t0 episode is actually a relapse from a recent infection, rather than a new infection that may generate hypnozoites, and (ii) exclude subjects with exceptionally high risk, as their fast infection dynamics might be heavily affected by new infections in addition to relapses. Here we impose a six-month malaria-free window before t0 as an inclusion criterion, but even longer windows may be required in temperate settings where late *P. vivax* relapses are common [2].

The present study has some limitations. First, malaria episodes diagnosed in pregnant women and controls were retrieved retrospectively from a case notification database. No blood samples were available for further confirmatory (e.g., molecular) diagnostic tests and compliance with the prescribed treatment was not evaluated. We assume that nearly all clinical malaria episodes diagnosed by microscopy and treated in cases and controls were retrieved. Accordingly, over 99% of clinical malaria cases diagnosed in Brazil are estimated to be entered in the electronic malaria notification database [41]. However, routine surveillance and case

notification overlook transient subpatent and asymptomatic *P. vivax* recurrences, which are very common in endemic settings across the Amazon [42]. Because antenatal malaria screening is not widely applied in our study site, nearly all malarial infections diagnosed and treated during pregnancy had been identified passively, when febrile women sought treatment in malaria outposts. This precludes any analysis of the burden of repeated asymptomatic parasite carriage in this population, which may lead to the formation of a large hypnozoite pool. Second, some repeated episodes in pregnant and control women may have been missed by our matching strategy in the absence of unique patient identifiers [23]. Nevertheless, we argue that this is unlikely to introduce a major bias in our comparisons, since similar proportions of mismatches are expected for cases and controls. Finally, we assume that pregnant women and non-pregnant controls are similarly exposed to infectious mosquito bites and similarly susceptible to malaria-related illness upon infection. This may not be the case, at least for *P. falciparum* malaria in Africa [28]. In fact, pregnant women appear to be more attractive to African malaria vectors [43], although not necessarily to mosquitoes in the Neotropics, where *P. vivax* predominates [28]. Moreover, pregnant women are more likely than non-pregnant women to develop clinical illness due to *P. falciparum* given the parasite's ability to massively sequester and multiply in the intervillous spaces of the placenta [29]. In contrast, *P. vivax* does not seem to sequester and accumulate in placental tissues [44, 45] and it remains uncertain whether increased risk of disease upon infection in pregnant women, compared to non-pregnant counterparts, also applies to *P. vivax*. Available evidence is conflicting, but there may be a slightly increased overall risk of *P. vivax* malaria during pregnancy, compared to non-pregnancy [28]. Nevertheless, the few studies that investigated the risk of *P. vivax* malaria in pregnancy did not attempt to distinguish between relapses (that are more common in pregnancy due to PQ-ineligibility) and new infections. Here we provide evidence to support the assumption that pregnant cases and non-pregnant controls have the same rate of new infections ($\hat{\beta}_n$) in our study population. We show that the common $\hat{\beta}_n$ parameter inferred from the time-to-the-next-event analysis (Fig 3) can also describe the rate of new infections in the time-to-the-previous-event starting six months *before* the baseline (S1 Text), when most "pregnant cases" were not yet pregnant. These findings imply that study participants experience the same average rate of new infections before and after the baseline, being little influenced by their pregnancy status that changed over time. Therefore, the extra burden of repeated *P. vivax* infections in pregnancy in our population is essentially due to PQ-preventable relapses and late recrudescences, rather than more frequent new infections.

We conclude that more widely prescribing post-treatment prophylaxis with weekly CQ in the management of vivax malaria in pregnancy may minimize the consequences of PQ ineligibility, by preventing parasite recurrences and averting adverse effects on mothers' and neonates' health.

## Supporting information

**S1 Checklist. STROBE Checklist.**
(PDF)

**S1 Data. Data.** Excel spreadsheet with time-to-event data from cases and controls.
(XLSX)

**S1 Text. Time-to-previous-episode.** Time-to-previous-episode analysis over 12 months of follow-up in PQ-untreated pregnant women and PQ-treated non-pregnant control experiencing a baseline vivax malaria episode.
(PDF)

## Acknowledgments

We thank Dr. Arnold Reynaldi (University of New South Wale, Kirby Institute) and Dr. Timothy Schlub (University of Sydney, School of Public Health) for helpful discussions.

## Author Contributions

**Conceptualization:** Rodrigo M. Corder, Miles P. Davenport, Marcelo U. Ferreira.

**Data curation:** Marcelo U. Ferreira.

**Formal analysis:** Rodrigo M. Corder, Antonio C. P. de Lima, David S. Khoury, Steffen S. Docken, Miles P. Davenport, Marcelo U. Ferreira.

**Funding acquisition:** Rodrigo M. Corder, Antonio C. P. de Lima, David S. Khoury, Miles P. Davenport, Marcelo U. Ferreira.

**Investigation:** Rodrigo M. Corder, Antonio C. P. de Lima, David S. Khoury, Steffen S. Docken, Miles P. Davenport, Marcelo U. Ferreira.

**Methodology:** Rodrigo M. Corder, Antonio C. P. de Lima, David S. Khoury, Steffen S. Docken, Miles P. Davenport, Marcelo U. Ferreira.

**Project administration:** Miles P. Davenport, Marcelo U. Ferreira.

**Software:** Rodrigo M. Corder, David S. Khoury, Steffen S. Docken.

**Supervision:** Miles P. Davenport, Marcelo U. Ferreira.

**Validation:** Rodrigo M. Corder, David S. Khoury, Steffen S. Docken.

**Visualization:** Rodrigo M. Corder, Antonio C. P. de Lima, David S. Khoury, Steffen S. Docken, Miles P. Davenport, Marcelo U. Ferreira.

**Writing – original draft:** Rodrigo M. Corder, Antonio C. P. de Lima, David S. Khoury, Steffen S. Docken, Miles P. Davenport, Marcelo U. Ferreira.

**Writing – review & editing:** Rodrigo M. Corder, Antonio C. P. de Lima, David S. Khoury, Steffen S. Docken, Miles P. Davenport, Marcelo U. Ferreira.

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
