## [Decision Letter · Decision Letter 0]

22 Apr 2020

Dear Mr. Corder,

Thank you very much for submitting your manuscript "Quantifying and preventing Plasmodium vivax recurrences in primaquine-untreated pregnant women: an observational and modeling study in Brazil" for consideration at PLOS Neglected Tropical Diseases. As with all papers reviewed by the journal, your manuscript was reviewed by members of the editorial board and by several independent reviewers. In light of the reviews (below this email), we would like to invite the resubmission of a significantly-revised version that takes into account the reviewers' comments. 

We cannot make any decision about publication until we have seen the revised manuscript and your response to the reviewers' comments. Your revised manuscript is also likely to be sent to reviewers for further evaluation.

Sincerely,

Lilian Lacerda Bueno, Ph.D.

Guest Editor

Genevieve Milon

Deputy Editor

Reviewer's Responses to Questions

**Key Review Criteria Required for Acceptance?**

**Methods**

-Are the objectives of the study clearly articulated with a clear testable hypothesis stated?

-Is the study design appropriate to address the stated objectives?

-Is the population clearly described and appropriate for the hypothesis being tested?

-Is the sample size sufficient to ensure adequate power to address the hypothesis being tested?

-Were correct statistical analysis used to support conclusions?

-Are there concerns about ethical or regulatory requirements being met?

Reviewer #1: In this study, the authors applied a mathematical model to quantify the extra burden of recrudescence attributable to PQ ineligibility in pregnant women. This study is very interesting and important since, although the prophylactic use of CQ is recommended, it is rarely prescribed in the study area. In this regard, the information provided by this study may help local authorities reinforce the need for prescription chloroquine as a prophylactic drug during pregnancy.

The objectives are clear, the study population and the criteria for inclusion or exclusion patients were well defined and the mathematical model used was adequate.

However, some issues need to be further clarified: 

1-Exponential smoothing, in which β rates and time dependency are entered, seems interesting to achieve a profile similar to the Klapan-Meier plot, which is used to describe the survival function for time-evolving data. However, it is not clear how to determine the proportion (pp ou pc), what would be a suitable value for it, if it is extracted exclusively from the plot and if it would vary with another model. 

2- I suggest that the authors better describe how β rates were obtained. I don't know if it's common knowledge for those in the field, but I would feel more comfortable if the authors explained how to get these β values and describe it in the text (line 298-300).

3- What was the criterion used to determine the size of the study population? It is not clear why the authors analyzed data from 1 January 2014 through 30 September 2016. What criteria for setting this date?

4- Considering that the sample size (pregnant and non-pregnant) was different (156 versus 316), do the authors believe that this could influence the results found?

Reviewer #2: The manuscript seems to be well written and the methods appear to be robust.

However, I have made the following observations:

 1. Exclusion criteria (6 months prior to baseline) seems to be theoretically sound but debatable practically. No effort to distinguish between recrudescence and new infection was made. Without proper molecular analysis, mere exclusion will not reveal the true picture. The authors thus should justify their study design and discuss their shortcomings in the discussion.

 2. Moreover, pregnancy is inherently an immuno-compromised state; comparison of recurrence or recrudescence in pregnant women vs. non-pregnant women is a definite bias.

 3. Since WHO already recommends IPT with SP during pregnancy in Africa, the same can be explored in South American countries like Brazil instead of advocating CQ prophylaxis.

**Results**

-Does the analysis presented match the analysis plan?

-Are the results clearly and completely presented?

-Are the figures (Tables, Images) of sufficient quality for clarity?

Reviewer #1: Overall the results have been presented very clearly and are well described in the text. However, 

- The legend of figure 5 is not very well described and I could not find enough information in the text either. I suggest that the authors clarify this part in the text and in the figure.

Reviewer #2: The results seems well presented and robust statistical analysis. However, the model simulations used in the study needs to be evaluated by an expert in this field and I would not like to comment on these.

**Conclusions**

-Are the conclusions supported by the data presented?

-Are the limitations of analysis clearly described?

-Do the authors discuss how these data can be helpful to advance our understanding of the topic under study?

-Is public health relevance addressed?

Reviewer #1: The conclusions reached with the study are of great relevance and provide important information about the importance of using chloroquine as a prophylactic method during pregnancy.

- If the use of a low dose of chloroquine as a prophylactic is recommended in Brazil, why is it rarely prescribed ? I suggest the authors comment on this in the text. Also, is this a feature of the local area of study? Would it be possible to apply this model to other areas of Brazil where chloroquine is prescribed correctly for pregnant women?

- The authors discuss the limitations of the study very well and in an impartial way (page 443). I would like the authors to discuss the fact that primaquine intake cannot be controlled in this type of study, and how incomplete treatment with this drug could influence the results found.

Reviewer #2: The conclusions are well written and supports the hypothesis.

**Editorial and Data Presentation Modifications?**

Reviewer #1: (No Response)

Reviewer #2: (No Response)

**Summary and General Comments**

Reviewer #1: (No Response)

Reviewer #2: (No Response)

PLOS authors have the option to publish the peer review history of their article (what does this mean?). If published, this will include your full peer review and any attached files.

Reviewer #1: No

Reviewer #2: No
---

## [Decision Letter · Decision Letter 1]

26 Jun 2020

Dear Mr. Corder,

We are pleased to inform you that your manuscript 'Quantifying and preventing Plasmodium vivax recurrences in primaquine-untreated pregnant women: an observational and modeling study in Brazil' has been provisionally accepted for publication in PLOS Neglected Tropical Diseases. We have attached a document with some suggested changes to your abstract to this decision letter, but those changes can be made during the production process that follows.

Best regards,

Lilian Lacerda Bueno, Ph.D.

Guest Editor

Genevieve Milon

Deputy Editor

Reviewer's Responses to Questions

Key Review Criteria Required for Acceptance?

Methods

-Are the objectives of the study clearly articulated with a clear testable hypothesis stated?

-Is the study design appropriate to address the stated objectives?

-Is the population clearly described and appropriate for the hypothesis being tested?

-Is the sample size sufficient to ensure adequate power to address the hypothesis being tested?

-Were correct statistical analysis used to support conclusions?

-Are there concerns about ethical or regulatory requirements being met?

Reviewer #1: (No Response)

Reviewer #2: I have reviewed the revised version of the manuscript and in my opinion the authors have satisfactorily addressed all my concerns. Though I am not an expert in simulation study, I am sure the 1st reviewer has looked into this.

Results

-Does the analysis presented match the analysis plan?

-Are the results clearly and completely presented?

-Are the figures (Tables, Images) of sufficient quality for clarity?

Reviewer #1: (No Response)

Reviewer #2: Figures and tables are of good quality as well as clarity.

Conclusions

-Are the conclusions supported by the data presented?

-Are the limitations of analysis clearly described?

-Do the authors discuss how these data can be helpful to advance our understanding of the topic under study?

-Is public health relevance addressed?

Reviewer #1: (No Response)

Reviewer #2: Both conclusion as well as limitations of the study is well written.

Editorial and Data Presentation Modifications?

Reviewer #1: (No Response)

Reviewer #2: Accept

Summary and General Comments

Reviewer #1: (No Response)

Reviewer #2: (No Response)

PLOS authors have the option to publish the peer review history of their article (what does this mean?). If published, this will include your full peer review and any attached files.

Do you want your identity to be public for this peer review?

 For information about this choice, including consent withdrawal, please see our Privacy Policy.

Reviewer #1: Yes: Anna Caroline Campos Aguiar

Reviewer #2: Yes: Dr Md Atique Ahmed

---

## [Editor Report · Acceptance letter]

23 Jul 2020

Dear Mr. Corder,

We are delighted to inform you that your manuscript, "Quantifying and preventing *Plasmodium vivax* recurrences in primaquine-untreated pregnant women: an observational and modeling study in Brazil," has been formally accepted for publication in PLOS Neglected Tropical Diseases.

Best regards,

Shaden Kamhawi

co-Editor-in-Chief

Paul Brindley

co-Editor-in-Chief
